

# Examination of a novel dietary fiber formulation on morphology and nutritional physiology of young male Sprague-Dawley rats fed a high fat diet

Milena Figueiredo de Sousa[1,2], Jingyu Ling[1], Eduardo Asquieri[2], Corrie Whisner[1] and Karen L. Sweazea[1,3]

[1] College of Health Solutions, Arizona State University, Phoenix, Arizona, United States
[2] School of Pharmacy, Universidade Federal de Goiás, Goiania, Brazil
[3] School of Life Sciences, Arizona State University, Tempe, AZ, United States

Corresponding authors
Milena Figueiredo de Sousa,
milenafnut@gmail.com
Karen L. Sweazea,
Karen.Sweazea@asu.edu

## ABSTRACT

Western diets are a public health concern as excess intake of simple sugars and fatty foods, and consequently low consumption of fruits and vegetables, can contribute to obesity and other chronic diseases such as diabetes mellitus, metabolic syndrome, cardiovascular diseases, and cancer. Due to the high prevalence of diseases related to Western diets, the objective of this study was to evaluate whether the inclusion of a novel fiber-rich complex could prevent high fat diet-induced weight gain, adiposity, hyperglycemia, dyslipidemia, and oxidative stress in young male Sprague-Dawley rats, *Rattus norvegicus*. The novel fiber complex contained a blend of bioactive ingredients: 27% flaxseed, 15.9% wheat bran, 14.8% wheat germ, 10% psyllium, 13.1% brewer's yeast, and 19.2% grapeseed flour. The study included 24 6-week-old rats divided into three groups that were fed either a control diet (C; standard rodent maintenance diet) containing fiber (3.8%g diet); high-fat diet (H) containing Solka Floc cellulose fiber (6.46%g diet); or high-fat diet in which 5% of the diet was replaced with the novel fiber complex (HF) (total fiber: 5%g fiber complex + 6.14%g Solka Floc). Rats in all diet groups gained significant weight during the 6-week feeding period ($p < 0.001$) consistent with normal growth. Whereas no differences were observed for blood lipids or beta-hydroxybutyrate, consumption of the H diet significantly increased adiposity ($p < 0.001$), liver triglycerides ($p < 0.001$), and fasting whole blood glucose concentrations ($p < 0.001$) in comparison to the C diet. These effects of high fat consumption were not prevented by the inclusion of the novel fiber complex in this experimental design.

## INTRODUCTION

Diets high in processed and ultra-processed foods, commonly found in Western societies, are a public health concern worldwide as these energy-dense diets are rich in sugars, fats, and salts but low in fiber-rich fruits, grains, seeds and vegetables (*Saklayen, 2018*). Inadequate fiber intake favors the emergence of chronic diseases such as metabolic syndrome and cardiovascular disease (*Jabbari et al., 2023*). Faced with this challenging

scenario, and growing public interest in healthy eating, functional foods fortified with bioactive ingredients that have purported health benefits (*e.g.*, vitamins, antioxidants, fibers) have become increasingly popular (*Rashininejad, 2024*). In fact, the functional food market in the United States alone was worth $56.4 billion (USD) in 2020 and has grown at a rate of 8.5% annually, on average (*Rashininejad, 2024*). Relevant to the present study, products enriched with dietary fiber, specifically, are increasingly popular and thought to improve health (*Tao et al., 2023*).

Dietary fibers can be defined as plant materials (*e.g.*, cellulose, pectin, lignin) that cannot be hydrolyzed by digestive enzymes during the digestive process. As such, the term "dietary fiber" is given to a set of non-digestible carbohydrate compounds present in plants with a structural function (*Khorasaniha et al., 2023*). Foods rich in dietary fiber are associated with a balanced diet and promote several health benefits (*Ito et al., 2023*; *Ramirez, Manuale & Yori, 2023*). For example, studies have shown that regular consumption of dietary fiber can help reduce chronic non-communicable diseases such as diabetes, hypertension, cardiovascular disease, and obesity (*Ito et al., 2023*; *Ramirez, Manuale & Yori, 2023*). Dietary fibers are noted for their ability to delay gastric emptying, increase motility of the large intestine, delay the absorption of glucose through the intestinal mucosa into the circulation, and reduce plasma cholesterol in humans (*Tariq et al., 2023*). In addition, fibers regulate the immune system and are fermented by intestinal bacteria that release short-chain fatty acids (SCFA) as a substrate, which can be used as a source of metabolic energy in various tissues (*Oshiro et al., 2022*; *Johnson, 2023*).

For many years, fibers were described and characterized according to their solubility (soluble and insoluble fibers). Dietary guidelines recommend adults consume 14 g of fiber per 1,000 calories. Current recommendations do not, however, predict the physiological effects of fibers based on differences in solubility, as both exert similar effects, but differ in their mechanisms (*Gill et al., 2021*). Fibers can also be divided according to their molecular weight with cellulose, hemicellulose, gums, mucilage, pectin, beta-glucans, resistant starch, and lignin considered high molecular weight. High molecular weight fibers are associated with health benefits as they can normalize bowel function *via* laxative or bulking effects and may lower cholesterol (*Stribling & Ibrahim, 2023*). Oligosaccharides and inulin, on the other hand, are considered low molecular weight prebiotic fibers that are fermented by gut microbiota to produce SCFA that play a role in gut homeostasis by regulating appetite, gut motility and barrier integrity, bacterial composition, as well as immune function, although rapid fermentation of low molecular weight fibers is associated with higher gas production (*Stribling & Ibrahim, 2023*).

Foods rich in soluble and insoluble fiber have been well-cited in the scientific literature for various beneficial effects in preclinical and clinical trials. Flaxseed is rich in soluble and insoluble fibers, antioxidants, and omega-3 fatty acids that help prevent osteoporosis, systemic inflammation, and cardiovascular diseases (*Aguilar et al., 2017*; *Kajla, Sharma & Sood, 2015*). Thus, flaxseeds may help reduce the risk of diseases associated with excess weight and hypertension (*Machado et al., 2015*). Similarly, male 6-week-old Sprague-Dawley rats, *Rattus norvegicus*, fed a high fat diet (1% and 15% higher cholesterol and lard, respectively) supplemented daily with a ≤ 1 kDa sized flaxseed peptide at an oral

dose of 200–800 mg/kg body mass had lower cholesterol and triglycerides in addition to reduced simple steatosis, compared to high fat diet alone (*Yuan et al., 2022*). Moreover, 1.2–2.4 g/day flaxseed powder has been shown to reduce blood pressure in deoxycorticosterone acetate (DOCA)-salt hypertensive male Wistar/ST rats (*Watanabe et al., 2020*).

From the industrial processing (*e.g.*, milling) of wheat, wheat bran can be captured and separated from the endosperm and germ layers of the kernels. Wheat bran is rich in dietary fiber and antioxidant compounds. It is considered an excellent source of dietary fiber, which makes up about 50% of its composition (*Li et al., 2023*). A study of male Sprague-Dawley rats fed a high-sucrose diet showed that 10–20%g diet wheat bran was effective at decreasing cholesterol whereas 5–20%g diet decreased lipids and glucose (*Ahmad & Takruri, 2015*). Similarly, 10%g or more wheat bran incorporated into the diet of male Sprague-Dawley rats was shown to decrease cholesterol and lipids (*Thannoun, 2005*). Moreover, administration of feruloyl oligosaccharides from wheat bran increased antioxidants while decreasing oxidative stress and glucose concentrations in type 1 diabetic rats (*Ou et al., 2007*) while carboxymethylated wheat bran fibers similarly improved glucose regulation in high fat diet and streptozotocin (STZ)-induced type 2 diabetic C57BL/6J mice, *Mus musculus* (*Li et al., 2021*). The consumption of wheat germ can also improve intestinal motility and reduce inflammation thereby exerting immunomodulatory effects in human subjects (*Hasanloei et al., 2021*).

Grape seed flour is a promising ingredient as it is a natural dietary source of antioxidants (*Özvural & Vural, 2011*). Studies have shown that the proanthocyanidins present in grape seed flour induce apoptosis and prevent metastasis of cultured breast and colon cancer cells (*Lutterodt et al., 2011*). In mice with high fat diet-induced obesity, grape seed flour has been shown to prevent obesity, increase adipose tissue thermogenesis (*Zhou et al., 2019*), improve insulin sensitivity, reduce cholesterol, and attenuate hepatic steatosis (*Seo et al., 2016*). Similarly, grape seed flour improves liver steatosis and lowers cholesterol as well as abdominal fat in hamsters with high fat diet-induced obesity (*Kim et al., 2014*).

The psyllium plant, *Plantago ovata*, originates from the eastern Mediterranean. The husks of the *Plantago ovata* seed (*Martellet et al., 2022*) are rich in gel-forming soluble dietary fiber and recent meta-analyses demonstrate the efficacy of psyllium fiber in treating symptoms related to metabolic syndrome and diabetes (*Gholami, Clark & Paknahad, 2024*; *Gholami & Paknahad, 2023*). Psyllium fiber supplementation (5%g diet) has also been shown to reduce epididymal and perirenal fat mass as well as triglycerides and total cholesterol in 7-week-old male Sprague-Dawley rats fed a 52%g high fat diet after 4 weeks (*Kang et al., 2007*). Supplementation with 5–10%g diet psyllium fiber for 21 days significantly reduced liver and serum cholesterol by increasing sterol biosynthesis in the liver of 30 and 90-day-old male Sprague-Dawley rats (*Arjmandi et al., 1997*; *Daggy et al., 1995*).

Brewer's yeast is a unicellular organism and a source of active peptides with antihypertensive, antihyperglycemic, antimicrobial, and rich antioxidant effects (*Hosseinzadeh et al., 2013*; *Oliveira et al., 2022*). Due to its valuable nutritional composition, it has been increasingly used as a supplement to food (*Amorim et al., 2016*).

Similarly, brewer's yeast biomass, a byproduct of brewing beer, was shown to exert anti-obesity and anti-diabetic effects in Sprague-Dawley rats including reduced lipids and insulin as well as increased liver antioxidants (*Chang & Kao, 2019*).

Given the encouraging results from studies examining individual bioactive dietary fibers and yeast, we examined the effects of a novel bioactive fiber complex at preventing high fat diet-induced dyslipidemia, hyperglycemia, and weight gain. While individual fibers exhibit beneficial effects on lipids, glucose, gut function, or oxidative stress, we predicted that the fiber complex would improve multiple symptoms in rats fed a high fat diet. Thus, this study aimed to understand the effects of fiber complex supplementation on glycemic, lipid, and morphological parameters in male rats. We hypothesized that the addition of a bioactive novel fiber complex to the diet of male Sprague-Dawley rats would prevent these characteristics of metabolic syndrome associated with a high fat diet.

## MATERIALS AND METHODS

All procedures were approved by the Arizona State University Institutional Animal Care and Use Committee (protocol 23-1971R) and complied with AVMA and NSF guidelines. Twenty-four healthy 6-week-old male Sprague-Dawley rats (*Rattus norvegicus*) weighing 141–165 g were used in this study (Inotiv, West Lafayette, IN, USA). Females were excluded as estrogen is protective from cardiovascular diseases and metabolic syndrome in young rats. Animals were housed in the Department of Animal Care Technology at Arizona State University. Animals were checked daily and veterinary staff were available to evaluate any animal showing signs of illness or distress as indicated by weight loss (≥10%), lethargy, ruffled coat, or respiratory distress, although no such instances occurred in the present study. From the moment the animals arrived in the facility, they were housed in pairs in 20.5″ × 10.5″ cages at an ambient temperature (74 ± 2 °F) and provided tunnels and Nyla bones for enrichment. Animals were exposed to a 12 h L:D cycle and room air was cycled at a standard rate of 10–15 air changes/hour. Rats were provided standard rodent diet (2018 Teklad Global 18% protein chow diet) and water *ad libitum* for the first few days after arrival. Animals were weighed 1 day after arrival to obtain baseline body mass (to the nearest g). Rats were randomly assigned after 3 days using the blocking method to one of three groups, in which a total of eight rats per group received one of the following diets from the initial day of the experiment: standard rodent chow (C); High-fat Diet (H); or H + Fiber complex (HF) for 6 weeks. Allocation of rats to each group and sample collections were not blinded. Measurements were collected at baseline (prior to switching diets), 3 and 6 weeks after diet allocation. Sample sizes were based on power analyses from prior studies of interventions used to assess and prevent high fat diet mediated hyperglycemia and symptoms of metabolic syndrome in rats (*Sweazea, Lekic & Walker, 2010*). Cages and food were replaced two times per week. After 6 weeks, the animals were euthanized (sodium pentobarbital, 200 mg/kg, i.p.) to collect organs, tissues, and draw cardiac blood using a 22-gauge needle for analyses of circulating factors as described below.

## Diets

As mentioned, all animals were provided standard plant-based rodent chow (Chow) maintenance diet (2018 Teklad; Inotiv, Indianapolis, IN, USA) upon arrival. According to the manufacturer, the standard rodent chow is comprised of (in kcal%) 58% carbohydrates (54% wheat, 40% corn, and the remainder of soybean meal), 24% protein (35% wheat, 34% corn, 26% soy products), and 18% fat (60% soybean products, 40% wheat and corn-derived). The chow diet also contained 3.8%g diet of fiber (79.5% neutral detergent fiber and 20.5% crude fiber). The macronutrient composition of the high fat diet with and without the fiber complex is shown in Table 1. The high fat diet was purchased from Research Diets Inc. (D12492; New Brunswick, NJ, USA) and its nutritional composition was provided by the manufacturer. For the preparation of the fiber complex, six raw bioactive materials rich in soluble and insoluble fibers were weighed in proportions described in Table 2, ground using a food processor (Osterizer Classic; Sunbeam-Oster Co., Inc, Fort Lauderdale, FL, USA), and stored at −20 °C. The HF diet was created by mixing 5% of the novel fiber complex and 95% high fat diet into pellets. The HF diet pellets were prepared fresh weekly and stored at −20 °C until use.

## Morphometrics

Body mass was measured weekly (to the nearest g) to assess changes in response to the diets. Nasoanal and tail length as well as abdominal circumference (immediately anterior to the hindleg) were measured using a flexible measuring tape (in cm) at the end of the trial. Following euthanasia, the epididymal fat pad was extracted from each animal and weighed to the nearest tenth of a gram using an analytic balance (Mettler Toledo, Greifensee, Switzerland) to assess adiposity between groups (*Sweazea, Lekic & Walker, 2010*). Liver as well as hearts were weighed to the nearest tenth of a gram for comparison between groups.

## Liver triacylglycerol concentrations

Liver triacylglycerol concentrations were measured according to published methods (*Jouihan, 2012*). Briefly, 100–300 mg liver samples were digested overnight at 55 °C in 350 μL ethanolic potassium hydroxide (one part 30% potassium hydroxide and two parts ethanol). The volume was adjusted to a total of 1 mL with 50% ethanol then centrifuged for 5 min at 13,000 rpm. The resulting supernatant was placed in a microcentrifuge tube and the volume adjusted to 1.2 mL with 50% ethanol. Magnesium chloride (215 μL 1 M $MgCl_2$) was added to a 200 μL aliquot, vortexed, and allowed to sit on ice for 10 min after which the solution was centrifuged at 13,000 rpm for 5 min. The resulting supernatant was placed in a fresh microcentrifuge tube. Liver free glycerol concentrations in the final supernatant were measured using a commercial kit according to the manufacturer's protocol (Sigma Aldrich, St. Louis, MO, USA). Triacylglycerol concentrations (mg/g tissue) were calculated from the liver free glycerol concentrations as follows: [glycerol] (mg/dl) * (10/30) * (415/200) * 0.012 (dL)/tissue mass (g).

**Table 1 Macronutrient profile of the high fat diet with and without fiber complex.**

| Ingredients (%g diet) | H (60%kcal high fat diet) | HF (95% H diet + 5% fiber complex) |
|---|---|---|
| **Total Proteins** | **26.23** | **24.92** |
| Casein, Lactic, 30 Mesh | 25.84 | 24.55 |
| Cystine, L | 0.39 | 0.37 |
| **Total Carbohydrates** | **25.56** | **24.29** |
| Lodex 10 | 16.15 | 15.35 |
| Sucrose, fine granulated | 9.41 | 8.94 |
| **Total Fats** | **34.89** | **33.15** |
| Lard | 31.66 | 30.08 |
| Soybean oil, USP | 3.23 | 3.07 |
| **Total Fibers** | **6.46** | **11.14** |
| Solka Floc, FCC200 | 6.46 | 6.14 |
| Novel fiber complex | 0 | 5 |
| **Added Vitamins/Minerals/Dye** | **6.86** | **6.52** |

Note:
Bold indicates total values for each category.

**Table 2 Raw materials used in the preparation of the fiber complex.**

| Raw material | Proportion (%) |
|---|---|
| Brown linseed (*i.e.*, flaxseed) | 27.0 |
| Brewer's yeast powder | 13.1 |
| Psyllium husk powder | 10.0 |
| Wheat bran | 15.9 |
| Wheat germ | 14.8 |
| Chardonnay grapeseed flour | 19.2 |

## Plasma biochemistry

Blood was collected from the caudal vein of all animals using a 26-gauge needle at weeks 3 and 6 for the analysis of fasting blood glucose and at week 6 to assess lipid profiles, beta-hydroxybutyrate, and oxidative stress. Rats were fasted by providing a small aliquot of food (3 g/rat at baseline and week 3 and 5 g/rat at week 6) the afternoon before (3 pm) fasting blood collections. The following morning, a dab of lidocaine containing cream was applied to the tail and blood samples (300–500 μL) were then collected from the caudal vein using a 26-gauge needle. Whole blood glucose concentrations were measured using a digital veterinary glucometer (AlphaTrak 2; Zoetis, Parsippany, NJ, USA). The lipid profile of whole blood was measured using an automatic biochemistry analyzer (CardioChek Professional; PTS Diagnostics, Whitestown, IN, USA) which provided data on total cholesterol, HDL cholesterol, and triglycerides.

Following euthanasia, blood was collected *via* cardiac puncture into heparinized and EDTA-containing vacutainers. Whole blood was centrifuged at 14,000 rpm at 4 °C and plasma was collected and stored at −80 °C for the measurement of triglycerides, free

glycerol, beta-hydroxybutyrate, and thiobarbituric acid reactive substances (TBARS, a marker of oxidized lipoproteins). Plasma triglycerides (total and true) and free glycerol were measured using a commercially available kit (Cat. No. TR0100; Sigma Aldrich, St. Louis, MO, USA). In this assay, free glycerol concentrations in the plasma are measured first, then triglycerides in the plasma are converted enzymatically using lipoprotein lipase to glycerol and free fatty acids to determine total triglyceride concentrations. True triglyceride concentrations thus reflect total triglycerides minus the free glycerol concentrations initially present in the samples. Beta-hydroxybutyrate (Cat. No. 700190; Cayman Chemical, Ann Arbor, MI, USA) and TBARS (Cat. No. 0801192; Zeptometrix, Buffalo, NY, USA) were analyzed according to manufacturer protocols using commercially available kits.

### Statistical analyses

The data are presented as means ± SEM. Statistical analyses were conducted using either two-way repeated measures ANOVA with diet and time as factors for data collected at multiple time points or one-way ANOVA for data collected at the conclusion of the study (SigmaPlot 14.0; Systat Software, Palo Alto, CA, USA). Data that were not normally distributed were log-transformed prior to statistical analyses (epididymal fat pad mass and plasma true triglycerides). Kruskal-Wallis one-way ANOVA on ranks were used to analyze data that were not normally distributed following log transformation. *Post-hoc* Student-Newman-Keuls analyses were used to compare data between and within groups. A significance level of ≤0.05 was considered statistically significant for all comparisons. Data for epididymal fat pad mass was missing for one rat in the HF group.

## RESULTS

### Morphometrics

Figure 1 shows the weekly body mass measurements of rats from each group. There were no significant differences in baseline body mass measurements between the rats in each group ($153 \pm 2.3$ g [C]; $156 \pm 1.6$ g [H]; and $156 \pm 2.1$ g [HF]). Rats in all diet groups gained significant weight over the 6-week feeding period (Time: $F_{6,126} = 1,148.618$, $p < 0.001$; power: 1.000) associated with typical growth. However, there were no significant differences in weight gain between the diet treatments ($F_{2,126} = 0.508$, $p = 0.609$) and no interactions between diet and time ($F_{12,126} = 1.160$, $p = 0.319$). Rats were weighed twice during the last week of the study. Shown in Table 3 are the body mass measurements just prior to euthanasia. The body mass of rats in each group at this time point measured (in g) $314 \pm 7.6$ [C], $324 \pm 7.5$ [H], and $331 \pm 8.6$ [HF] and were not significantly different ($F_{2,21} = 1.162$, $p = 0.332$).

Animal morphology measures at week 6 are shown in Table 3. There were no significant differences in the size of rats fed each diet as measured by tail ($F_{2,21} = 0.525$, $p = 0.599$) or nasoanal length ($F_{2,21} = 0.135$, $p = 0.875$). Consistent with the body mass data, abdominal circumference also showed no significant variation across groups ($F_{2,21} = 0.766$, $p = 0.477$). However, rats fed high fat diet had significantly more adiposity as compared to rats fed the standard rodent chow as measured by epididymal fat pad mass ($F_{2,20} = 15.623$, $p < 0.001$;

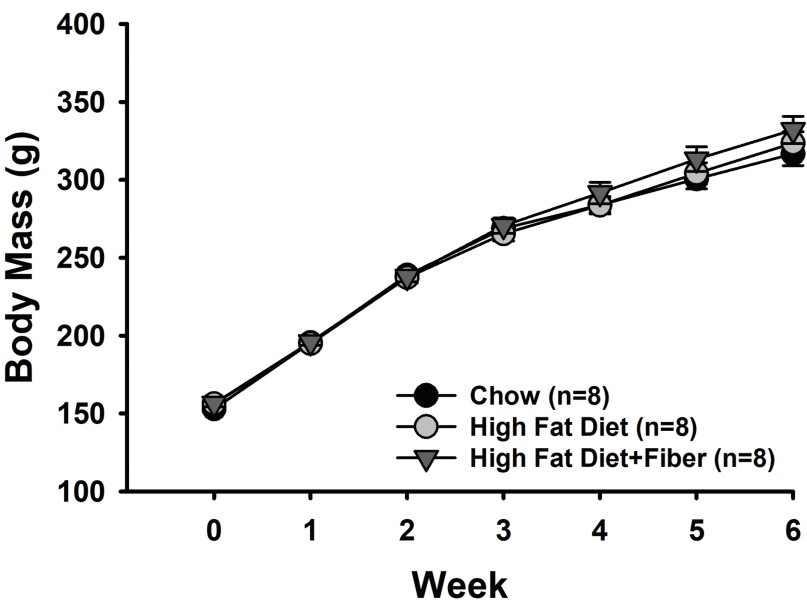

**Figure 1 Weekly body mass measurements.** The alteration in body mass from weeks 0 to 6 is presented as the mean ± SEM. A two-way repeated measures ANOVA was conducted with diet and time as factors to analyze the data. Throughout the 6-week diet period, all rats experienced weight gain (Time: $p < 0.001$). No significant difference were observed in weight gain among the various diet treatments (Diet: $p = 0.609$), and no interactions were found between diet and time ($p = 0.319$). Each group consisted of $n = 8$ male Sprague-Dawley rats.

**Table 3 Animal morphology after 6 weeks of each diet.**

|  | Chow diet | High fat diet (H) | H + Fiber (HF) | Statistics |
|---|---|---|---|---|
| **Body mass (g)** | 314 ± 8 ($n = 8$) | 324 ± 7 ($n = 8$) | 331 ± 9 ($n = 8$) | $F_{2,21} = 1.162$, $p = 0.332$ |
| **Abdominal circumference (cm)** | 15.3 ± 0.2 ($n = 8$) | 15.6 ± 0.2 ($n = 8$) | 15.7 ± 0.2 ($n = 8$) | $F_{2,21} = 0.766$, $p = 0.477$ |
| **Tail length (cm)** | 20.2 ± 0.2 ($n = 8$) | 20.3 ± 0.3 ($n = 8$) | 20.5 ± 0.3 ($n = 8$) | $F_{2,21} = 0.525$, $p = 0.599$ |
| **Naso-anal length (cm)** | 21.3 ± 0.2 ($n = 8$) | 21.4 ± 0.2 ($n = 8$) | 21.3 ± 0.2 ($n = 8$) | $F_{2,21} = 0.135$, $p = 0.875$ |
| **Epididymal fat pad mass (g)** | 2.7 ± 0.2[a] ($n = 8$) | 4.4 ± 0.3[b] ($n = 8$) | 4.6 ± 0.5[b] ($n = 7$) | $F_{2,20} = 15.623$, $p < 0.001$ |
| **Liver mass (g)** | 12.1 ± 0.6 ($n = 8$) | 10.7 ± 0.4 ($n = 8$) | 11.1 ± 0.4 ($n = 8$) | $F_{2,21} = 2.760$, $p = 0.086$ |
| **Heart mass (g)** | 1.1 ± 0.03 ($n = 8$) | 1.1 ± 0.02 ($n = 8$) | 1.2 ± 0.06 ($n = 8$) | $H = 3.635$, df = 2, $p = 0.162$ |

**Note:**
Data expressed as mean ± SEM, analyzed by one-way ANOVA. Data for heart mass were not normally distributed and were analyzed by Kruskal-Walllis One-way ANOVA on Ranks with Dunn's *post-hoc* analyses. Different lowercase letters indicate significant differences between groups (Dunn's Method, $p = 0.005$).

power: 0.996). Notably, in this experimental design, the fiber complex did not prevent high fat diet-induced increases in epididymal fat pad mass ($p = 0.831$ H v HF). Neither liver nor heart mass were significantly different between groups ($p = 0.086$ and $0.162$, respectively).

## Biochemical parameters

Baseline whole blood glucose concentration prior to the start of the dietary treatments was 132.9 ± 5.8 mg/dL. Fasting whole blood glucose concentrations measured at weeks 3 and 6 are shown in Fig. 2. Two-way repeated measures analysis of variance indicated that diet significantly affected fasting whole blood glucose concentrations at weeks 3 and 6 ($F_{2,21} = 13.334$, $p < 0.001$; power: 0.990), whereas neither time ($F_{1,21} = 0.736$, $p = 0.401$) nor

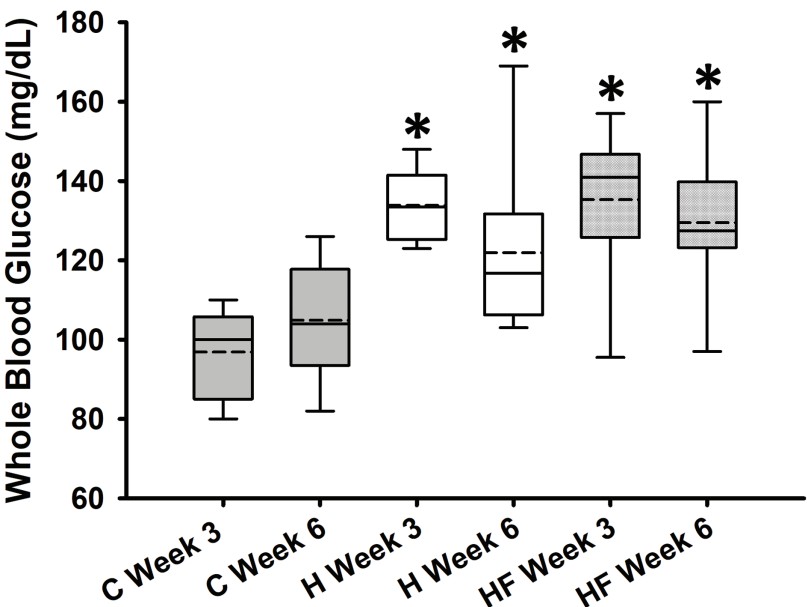

**Figure 2 Whole blood glucose concentrations after 3 and 6 weeks following each diet.** Data are expressed as means ± SEM. Boxplots show the interquartile range with the solid horizontal line indicating the median and the dashed line indicating the mean. The whiskers depict the minimum and maximum values. Data were analyzed by two-way repeated measures ANOVA with diet and time as factors (Diet: $p < 0.001$; Time: 0.401; Diet × Time: $p = 0.113$). Student-Newman-Keuls *post-hoc* analyses revealed that animals fed either high fat diet (H) or high fat diet + fiber (HF) had significantly higher whole blood glucose concentrations as compared to animals fed the standard rodent chow (C), *$p < 0.001$. In this experimental design, fiber supplementation did not prevent high fat diet-induced hyperglycemia ($p = 0.503$). Each dietary group comprised $n = 8$ male Sprague-Dawley rats.

interactions between diet and time ($F_{2,21} = 2.422$, $p = 0.113$) were significant. Student-Newman-Keuls *post-hoc* analysis revealed that both H and HF animals had significantly higher fasting blood glucose concentrations compared to rats fed the standard rodent chow at both time points ($p < 0.001$). However, in this experimental design, the fiber complex did not prevent high fat diet-induced fasting hyperglycemia ($p = 0.503$ H *vs.* HF).

For most animals, fasting total and LDL cholesterol concentrations were below the detection limit of the meter, thus data are not reported for these variables. Mean fasting whole blood HDL cholesterol and triglyceride concentrations for all animals at baseline were 66.1 ± 2.3 mg/dL and 77.8 ± 7.1 mg/dL, respectively. Fasting whole blood HDL cholesterol and triglyceride concentrations at week 6 are shown in Table 4. There were no significant differences between groups in whole blood fasting HDL ($F_{2,21} = 1.379$, $p = 0.274$) or triglycerides ($F_{2,21} = 0.482$, $p = 0.624$) after 6 weeks of each dietary treatment. Similarly, plasma true triglycerides ($F_{2,21} = 1.888$, $p = 0.176$) and free glycerol ($F_{2,21} = 0.657$, $p = 0.529$) were not significantly different between groups. The triacylglycerol content in the liver, shown in Fig. 3, was significantly different between the high-fat groups compared to the chow diet ($F_{2,20} = 15.349$, $p < 0.001$; power: 0.996) with animals fed a high fat diet developing hepatic steatosis that, in this experimental design, was not prevented with the fiber complex (H v HF $p = 0.468$). The levels of plasma

**Table 4 Lipid profile of rats after 6 weeks of each diet.**

| | Chow ($n = 8$) | High fat ($n = 8$) | High fat + Fiber complex ($n = 8$) | Statistics |
|---|---|---|---|---|
| **Fasting whole blood HDL cholesterol (mg/dL)** | | | | |
| | 55.6 ± 1.81 | 61.3 ± 2.65 | 59.4 ± 2.74 | $F_{2,21} = 1.379$, $p = 0.274$ |
| **Fasting whole blood triglycerides (mg/dL)** | | | | |
| | 74.0 ± 3.70 | 68.1 ± 3.00 | 71.0 ± 5.57 | $F_{2,21} = 0.482$, $p = 0.624$ |
| **Plasma total triglycerides (mg/dL)** | | | | |
| | 45.5 ± 9.85 | 59.6 ± 4.74 | 53.2 ± 8.27 | $F_{2,21} = 1.770$, $p = 0.195$ |
| **Plasma true triglycerides (mg/dL)** | | | | |
| | 37.3 ± 8.88 | 49.2 ± 3.99 | 43.5 ± 6.57 | $F_{2,21} = 1.888$, $p = 0.176$ |
| **Plasma free glycerol (mg/dL)** | | | | |
| | 8.17 ± 1.03 | 10.4 ± 1.03 | 9.70 ± 1.95 | $F_{2,21} = 0.657$, $p = 0.529$ |
| **Plasma beta-hydroxybutyrate (mM/L)** | | | | |
| | 0.21 ± 0.03 | 0.29 ± 0.03 | 0.27 ± 0.03 | $F_{2,21} = 1.838$, $p = 0.184$ |

Note:
Data expressed as mean ± SEM and analyzed by One-way ANOVA. True triglycerides = Total triglycerides – Free glycerol.

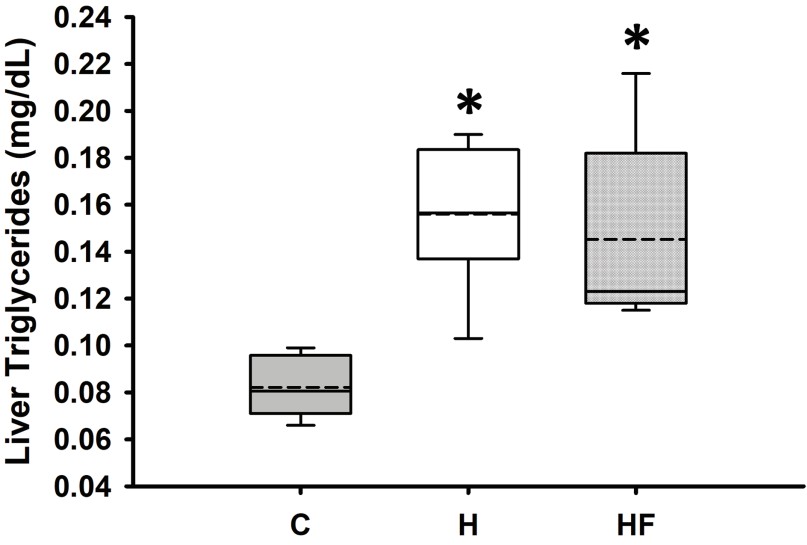

**Figure 3 Boxplot showing liver triglyceride concentrations 6 weeks after each diet.** Data are expressed as means ± SEM. Data were analyzed by one-way ANOVA and Student-Newman-Keuls *post-hoc* analyses. Boxplots show the interquartile range with the solid horizontal line indicating the median and the dashed line indicating the mean. The whiskers depict the minimum and maximum values. *$p < 0.001$ *vs.* standard rodent Chow.

beta-hydroxybutyrate (BHB) and TBARS measured in animals from each group did not demonstrate significant differences at the end of the study (Table 4 and Fig. 4, respectively).

## DISCUSSION

Contrary to the hypothesis, the novel fiber complex did not prevent hyperglycemia or hepatic steatosis in rats fed a high fat diet in this experimental design. The efficacy of

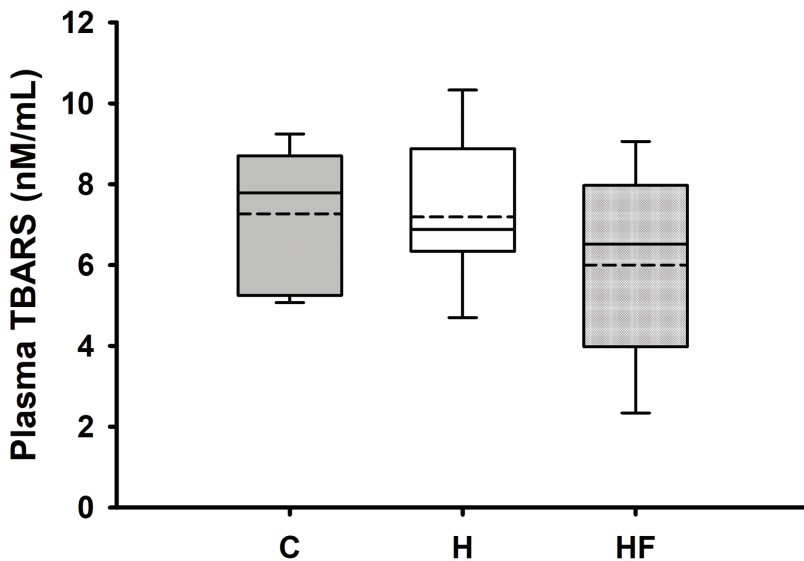

**Figure 4 Boxplot showing plasma oxidized lipoproteins (TBARS) 6 weeks after each diet.** Data are expressed as means ± SEM. Boxplots show the interquartile range with the solid horizontal line indicating the median and the dashed line indicating the mean. The whiskers depict the minimum and maximum values. Data were analyzed by one-way ANOVA and Student-Newman-Keuls *post-hoc* analyses. No significant differences in TBARS were observed between groups ($F2, 18 = 0.931$, $p = 0.412$).

dietary fibers depends on several factors, such as fiber type, particle size, solubility, surface area, viscosity, dose and duration. Sensory studies with rats demonstrate the animals' preference for ingesting fat, which is also common in humans due to its high palatability. An increase in intake reflects higher daily energy consumption (*Hariri & Thibault, 2010*). Regular consumption of high-calorie low-fiber diets contributes to metabolic syndrome and obesity, which are non-communicable diseases considered to be a public health concern as they contribute to high rates of morbidity and mortality. While obesity is defined as abnormal or excessive accumulation of body fat, metabolic syndrome is diagnosed when a patient exhibits at least three of the following symptoms: high fasting blood glucose or triglycerides, low HDL cholesterol, hypertension, or central obesity (*Estanyol-Torres et al., 2023*; *Yustisia et al., 2022*). Although obesity is commonly seen in people with metabolic syndrome, it is not a requirement for diagnosis. In fact, a study showed that 17% of individuals with normal body mass index had metabolic syndrome (*Suliga, Koziel & Gluszek, 2016*). Likewise, *Coelho et al. (2021)* noted that individuals with normal weight had a similar risk for developing metabolic syndrome when compared to obese individuals.

In a prior study, male Sprague-Dawley rats were fed either a diet high in saturated fats, sucrose, or a control diet for 6 weeks. The results demonstrated that the high fat and high sucrose groups gained more weight and epididymal fat mass compared to the chow diet (*Sweazea, Lekic & Walker, 2010*). Similarly, the results obtained in this study showed weight gain patterns consistent with normal growth and development during the present 6-week study, although rats fed the high fat diet gained significantly more epididymal fat

mass compared to those in the chow group. The fiber complex did not prevent high-fat diet-induced increases in adiposity in this experimental design. According to *Bastías-Perez, Serra & Herrero (2020)*, Sprague-Dawley rats gain weight easily on a high-fat diet. However, consistent with the current findings, the authors note that there is a wide variation in weight gain among rats, with weight gain appearing both in groups consuming high and low-fat diets. In fact, a study found that feeding young male Sprague-Dawley rats a diet with very little fat resulted in weight gain, compared to those fed control or high fat diets, which was attributed to increased lean mass (*Harris, 1991*). *Mateos-Aparicio et al. (2020)* evaluated 24 rats supplemented with fiber from an apple by-product until it comprised 20%g of their dietary fiber for 5 weeks. Corroborating the results of the present study, they did not find a significant difference in body mass or organ mass (liver and heart) between the control group and the group with fiber supplementation. In contrast, another study found that rats fed a high-fat diet for 8 weeks supplemented with fiber from black currant and strawberry (66.5 and 51.3%g) had decreased body weight and reduced epididymal fat mass (*Jurgoński et al., 2016*). Similarly, *Rotimi et al. (2012)* investigated the effects of high-fiber diets in obese rats and found the diets significantly reduced ($p < 0.05$) weight gain. These variations in weight gain and adiposity responses are similar to heterogenous responses observed in the development of human obesity. Although it is worth noting that variations in dietary fiber types, doses, and treatment lengths likely contribute to variations in outcomes.

High fat intake is also associated with fatty liver disease. A study of male Wistar rats observed that triglycerides as well as the ratio of visceral fat and liver mass to body mass were higher in the group fed a high-fat diet than in the control group fed standard chow, similar to the results of this study (*Liu et al., 2016*). The fiber complex did not prevent liver steatosis in the current experimental design. *Kanazawa et al. (2008)* studied four groups of male Sprague-Dawley rats fed a 60% sucrose diet (control) or one of three diets containing corn starch (CS), Benimaru potato starch (BM), or Hokkaikogane potato starch (HK). In the BM and HK diet groups, liver triglyceride levels decreased compared with the control and CS groups. Differing from this study, *Jiang et al. (2021)* evaluated the effects of dietary egg white protein hydrolysate on improving orotic acid-induced fatty liver in male Sprague-Dawley rats. After 14 days of feeding, no significant differences were found in initial and final body weights between the four groups. However, the egg white protein significantly reduced liver triglyceride levels. From a physiological point of view, a high-fat diet favors an increase in hepatic lipids. Excess free fatty acids in hepatocytes contribute to lipogenesis, inhibition of lipolysis, and increased lipid mass in the organ. Fatty acids in the liver may come from plasma non-esterified fatty acids, mainly derived from adipose tissue lipolysis; lipogenesis, mainly from glucose; and dietary free fatty acids in the form of chylomicrons. In the liver, such acids are oxidized by mitochondrial beta-oxidation or used to form triglycerides. The latter are exported to the bloodstream or accumulate in lipid concentrates in hepatocytes (*Malagó-Jr et al., 2021*; *Da Rocha et al., 2023*). The lipid profile of the animals in each dietary group showed no significant differences in the present study. However, further studies are warranted evaluating the efficacy of fiber complex supplementation in the prevention of high fat diet-induced dyslipidemia over a longer

duration study, since high fat diets may take longer to induce dyslipidemia in some rat strains, which may have been a limiting factor in the present work. Similar to the present findings, *Li et al. (2020)* evaluated rats supplemented with barley fiber and did not observe positive effects on serum levels of total cholesterol or HDL. However, *Nandi, Sengupta & Ghosh (2019)* evaluated a group of five adult rats for only 32 days, supplemented with fibers extracted from flaxseed, rice bran, and sesame husk and found a reduction in total and LDL cholesterol as well as triglyceride levels whereas HDL cholesterol was increased. This is contradictory to the findings from the present study of adolescent rats.

Following the 6-week study, animals in both the high-fat (H) and high-fat + fiber (HF) groups developed hyperglycemia compared to the control group (with no significant difference between the H and HF groups). Thus, in this experimental design, the fiber complex did not prevent hyperglycemia induced by a 6-week high fat diet. These findings are similar to a study aimed at assessing diets that induce insulin resistance and impair glucose metabolism in rats as a potential model for studying type 2 diabetes. Male Sprague Dawley rats were divided into three groups and fed control, high-fat, or high-fructose diets for 3 months. Surprisingly, all groups showed similar weight gain, glucose tolerance, insulin levels, and muscle glycogen synthesis, indicating that the rats adapted to the diets without developing insulin resistance or impaired glucose tolerance (*Stark, Timar & Madar, 2000*). In contrast, a recent study carried out by *Ge et al. (2023)* examining rats supplemented with insoluble fiber demonstrated benefits such as promoting liver health, reducing lipid deposits, and a positive impact on glycolipid metabolism. In a similar study, consumption of barley fiber from the Tibetan highlands by 60 rats for 15 weeks significantly reduced weight gain and dyslipidemia, improved glucose tolerance, and increased SCFA in the feces of mice (*Gan et al., 2023*). In contrast, results from the present study show that the fiber complex was not effective at preventing increased adiposity or hyperglycemia in rats fed a high fat diet. In light of the present findings and conflicting outcomes in the literature, it is not yet clear whether (or which) various dietary fibers provide beneficial roles in reducing adiposity. Another limitation is the potential for variation in the individual response to each type of fiber, formulation, and experimental design (*Wen et al., 2023*).

Beta-hydroxybutyrate (BHB) is an essential ketone body synthesized in the liver from fatty acids, serving as a crucial energy source during periods of low glucose availability, such as during ketogenic diets characterized by food deprivation and high-fat intake (*Newman & Verdin, 2017*). Although animals fed each of the high fat diets used in this study developed hyperglycemia, BHB were not significantly different between groups. Thiobarbituric acid reactive substances are commonly used to estimate lipid peroxidation, a marker of oxidative stress. Although elevated plasma and tissue TBARS have been reported in prior studies of rats fed a high fat diet for 6–12 weeks (*Elmarakby & Imig, 2009*; *Sweazea, Lekic & Walker, 2010*; *Tian et al., 2011*), the levels of TBARS measured in animals from each dietary group in the present study showed no significant differences after the 6-week intervention.

Limitations of the present study include small sample sizes, short duration feeding protocol, and lack of fiber-depleted chow and high fat diets. In fact, animals in the high fat

group consumed more fiber per gram of diet as compared to the standard chow diet (6.5 *vs.* 3.8%g diet, respectively). Thus, the addition of 5% bioactive dietary fibers to the high fat diet may not have been sufficient to prevent high fat diet induced adiposity and hyperglycemia. A meta-analysis of clinical trials found that the risk for developing metabolic syndrome is lower in people consuming the highest amounts of dietary fiber (*Chen et al., 2017*). However, while the authors observed an inverse relationship between fiber intake and metabolic syndrome risk, the relationship was not significant prompting a call for additional cohort studies to examine this putative link (*Chen et al., 2017*). Another limitation of the current experimental design may be the rapid switch from the chow to high fat diet with or without dietary fiber as such a switch may alter the gut microbiome and gastrointestinal function. We attempted to mitigate this by waiting 3–6 weeks after switching their diet to collect samples. As the animals were pair-housed for the duration of the study, as per AVMA guidelines, it was not possible to measure individual food intake or the activity of animals.

## CONCLUSIONS

While high fat intake resulted in increased adiposity and hyperglycemia, the addition of a 5% bioactive fiber complex was not effective, in the present experimental design, at preventing these deleterious effects of a high fat diet. The lack of overall weight gain combined with increased epididymal fat and hyperglycemia in rats consuming either high fat diet are similar to recent studies of humans whose body mass index fell within the normal range but had visceral obesity and fasting hyperglycemia (*Suliga, Koziel & Gluszek, 2016*). Moreover, while a meta-analysis of clinical trials observed an inverse relationship between fiber intake and metabolic syndrome risk, the relationship was not significant prompting a call for additional cohort studies (*Chen et al., 2017*). As such, although the present study did not support the addition of this novel fiber complex to prevent complications associated with short term high fat intake, additional studies that consider longer treatment periods or higher doses are recommended. In addition, studies that incorporate different dietary fibers and formulations are warranted to assess the most effective combinations for preventing or treating specific metabolic conditions. Our findings also raise the possibility that dietary fiber supplementation alone may not be sufficient to prevent the effects of continued consumption of such a high fat diet as that used in the present study. For this reason, experimental designs should consider other interventions to mitigate or prevent metabolic syndrome that compare dietary supplementation with dietary changes (*e.g.*, calorie reduction, low fat diet, intermittent fasting, *etc.*) and/or physical activity.

### Funding

This research was funded by the Arizona State University College of Health Solutions. Coordination for the Improvement of Higher Education Personnel (CAPES) provided funding to researcher Milena Figueiredo de Sousa during her exchange at Arizona State

University (Arizona-USA). Funds were additionally provided by the Graduate and Professional Student Association at Arizona State University (Jingyu Ling). The funders had no role in study design, data collection and analysis, decision to publish, or preparation of the manuscript.

## Grant Disclosures

The following grant information was disclosed by the authors:
Arizona State University College of Health Solutions.
Coordination for the Improvement of Higher Education Personnel (CAPES).
Arizona State University.

## Competing Interests

The authors declare that they have no competing interests.

## Author Contributions

- Milena Figueiredo de Sousa conceived and designed the experiments, performed the experiments, analyzed the data, prepared figures and/or tables, authored or reviewed drafts of the article, and approved the final draft.
- Jingyu Ling performed the experiments, analyzed the data, prepared figures and/or tables, authored or reviewed drafts of the article, and approved the final draft.
- Eduardo Asquieri conceived and designed the experiments, authored or reviewed drafts of the article, and approved the final draft.
- Corrie Whisner conceived and designed the experiments, authored or reviewed drafts of the article, and approved the final draft.
- Karen L. Sweazea conceived and designed the experiments, performed the experiments, analyzed the data, prepared figures and/or tables, authored or reviewed drafts of the article, and approved the final draft.

## Animal Ethics

The following information was supplied relating to ethical approvals (*i.e.*, approving body and any reference numbers):

All procedures were approved by the Arizona State University Institutional Animal Care and Use Committee and complied with AVMA and NSF guidelines (23-1971R).

## Data Availability

Raw data are available in the Supplemental Files.

## Supplemental Information

Supplemental information for this article can be found online at http://dx.doi.org/10.7717/peerj.19029#supplemental-information.

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
