# Peer review of "Examination of a novel dietary fiber formulation on morphology and nutritional physiology of young male Sprague-Dawley rats fed a high fat diet"

_PeerJ, doi:10.7717/peerj.19029_

## Round 0.1 · original submission · Major Revisions

Dear authors,

Manuscript titled "Examination of a novel dietary fiber formulation on morphology and nutritional physiology of young male Sprague-Dawley rats fed a high fat diet" that you submitted to PeerJ has been reviewed.

The reviewer(s) have suggested that some important points must be clarified and have requested substantial changes to be made in the manuscript. Therefore, I invite you to respond to the reviewer(s)' comments and revise your manuscript. The reviewer(s) comments are included at the end of this letter.

Please ensure that all review, editorial, and staff comments are addressed in a response letter and that any edits or clarifications mentioned in the letter are also inserted into the revised manuscript where appropriate.

Reviewer 1 ·

Basic reporting

.

Experimental design

.

Validity of the findings

.

Additional comments

I only read the abstract. I do believe that reading the entire manuscript wouldn't change my decision.
The novel fiber formula had no effect on improving nutritional status of the rats.
Negative results, unless under unusual circumstances, are seldom published.
My recommendation is to reject.

Something that might be tried is the effect of the novel fiber on the composition of the microbiota

·

Basic reporting

I have reviewed the manuscript titled "Examination of a novel dietary ûber formulation on morphology and nutritional physiology of young male Sprague-Dawley rats fed a high-fat diet." The study addresses an important topic and is generally well-structured, with clear aims and comprehensive methodology.
The manuscript follows a logical structure, clearly presenting the hypothesis, methodology, and results. The well-organized sections make the paper easy to follow and scientifically rigorous.
The study introduces a unique dietary fiber complex combining several functional ingredients, contributing new insights into nutritional science, particularly in high-fat diets.
The research was conducted in full compliance with ethical guidelines approved by the Institutional Animal Care and Use Committee (IACUC). Animal welfare was prioritized, and humane euthanasia procedures were used.
The tables, figures, and charts are well-organized and present data. The statistical analyses are sound and appropriate for the study's objectives, making the results easily interpretable.
I would recommend that please conduct a final proofread to catch minor grammatical errors or awkward phrasing.

Experimental design

Authors, please ensure that’s

The study effectively examines the short-term effects of the novel fiber complex.
The authors suggest that future studies consider extending the study duration beyond six weeks and experimenting with higher or varied fiber, which could reveal additional, long-term health benefits and clarify dose-dependent effects.
The comparison between high-fat and fiber-supplemented groups is valuable.
Including a fiber-depleted control group could provide additional insights into how fiber alone contributes to metabolic health, improving the interpretability of results.
The statistical methods used are appropriate and robust for the data presented.
Including effect sizes alongside p-values would help us better understand the practical significance of the findings, making the conclusions more impactful for future research.

Validity of the findings

The discussion critically reflects on the results, acknowledging the study's limitations while providing well-grounded conclusions. The authors appropriately recognize that the fiber complex did not prevent certain metabolic conditions but suggest reasonable areas for future investigation.
Delving deeper into the mechanisms by which individual fibers work and providing a more detailed comparison with conflicting evidence from prior studies further enhance the depth of the discussion.
The study adds valuable data to dietary fiber research, particularly in understanding how fiber formulations may impact metabolic health in the context of high-fat diets.
A more explicit discussion of how the results could inform future human studies or dietary guidelines would enhance the research's practical relevance.

Additional comments

The manuscript is written in professional language and is accessible to a broad scientific audience.
Simplifying some of the more complex sentences in the discussion and adding a concise summary of critical results would improve readability for a wider audience.

Reviewer 3 ·

Basic reporting

This is an interesting and useful nutritional study of rats when provided with three diet types. The study focuses on the effects of a high calorie, easily digestible diet, and the potential remedial effects of fibre in diets. The work is well supported with a wide range of relevant references, which are formatted to a consistent style. This provides a good overall understanding of the wider field of high-calorie diets - though a bit of care is needed with regards to terminology and definitions. For example, the use of the term 'poor quality diet' is used in the work: to many readers, this could be interpreted as a high fiber, low-calorie diet, because it is poor in macronutrients.
This said, the work is professionally formatted and raw data have been provided, along with a copy of the ARRIVE guidelines that demonstrate how the guidelines have been addressed in the work. The work is largely self-contained, except for a few points in the methods, where some aspects of the methods are detailed in other papers. For the sake of repeatability, there needs to be some clearer explanations of the actual methods of the study.

Experimental design

The work sites well withing the wider scope of the remit of PeerJ. The authors have provided a clear explanation of the research question and the work largely speaking is well formatted to address the point. However, the addition of fiber was relatively limited - more fiber may have resulted in greater effects for the rats. This should be explored in a bit more detail in the work.
It looks as if diet has been adjusted overnight from one diet to the experimental diet? If this is the case, there is likely some gastrointestinal impacts and microbiome changes for the rats: this needs to be explored in a bit more detail in the discussion. Similarly, effects of sample size and its ramifications should be considered in a bit more detail.

Validity of the findings

The work provides some interesting information in terms of obesity and fiber, though it is limited by the single fiber cohort, and a relatively small sample size. These effects, along with the diet change (without an acclimatisation period) need to be explored in the work in a bit more detail. This will result in a more holistic discussion of the effects of fiber and fat on rat health.
Please also see the point pertaining to correction factors for the statistical analysis: were any applied?

Additional comments

This is an interesting study, and I look forward to seeing further developments for this manuscript.

Annotated reviews are not available for download in order to protect the identity of reviewers who chose to remain anonymous.

---

## Round 0.2 · accepted · Accept

Dear Author,

Congratulations! After your diligent work addressing the reviewers' comments, I am pleased to inform you that your manuscript has been accepted for publication in PeerJ. This version is more concise and formal, enhancing clarity and flow.

·

Basic reporting

The authors have addressed all the comments and suggestions from the initial review, resulting in a significantly improved manuscript. The text is now polished, with grammatical errors and awkward phrasing corrected. Complex sentences have been simplified, and a concise summary of key findings has been added, improving readability and accessibility to a broader audience. Figures and tables are clear, well-organized, and effectively support the data.

The study design remains robust, with the rationale for the six-week duration and the absence of a fiber-depleted control group now well-explained. The statistical analyses are appropriate, and while effect sizes were not included, the authors have acknowledged this and highlighted its importance for future studies. The discussion has been enriched by exploring the mechanisms of the fiber formulation and comparing findings with prior research, providing greater depth and clarity.

The authors have connected their findings to human dietary interventions, increasing the practical relevance of the study. Suggestions for future work, such as exploring dose-dependent effects and conducting long-term human studies, provide a clear direction for further research.

Experimental design

All the recommendations have been rectified.

Validity of the findings

All the recommendations have been rectified.

Reviewer 3 ·

Basic reporting

Dear Authors,
Thank you for providing a clear response letter documenting the edits to the work, alongside the tracked changes manuscript. The original concerns have now been well addressed.

Experimental design

The experimental design is clearly explained and the edits have provided clearer information with regards to methodological design. There is sufficient information to allow replication of the study.

Validity of the findings

The results are non- significant, but this is not a problem as non significance is still a useful study finding in and of itself.

Additional comments

Thank you for providing a clear and well-considered set of revisions for your work.